# Depth Map Prediction from a Single Image using a Multi-Scale Deep Network

**David Eigen**
deigen@cs.nyu.edu

**Christian Puhrsch**
cpuhrsch@nyu.edu

**Rob Fergus**
fergus@cs.nyu.edu

Dept. of Computer Science, Courant Institute, New York University

## Abstract

Predicting depth is an essential component in understanding the 3D geometry of a scene. While for stereo images local correspondence suffices for estimation, finding depth relations from a *single image* is less straightforward, requiring integration of both global and local information from various cues. Moreover, the task is inherently ambiguous, with a large source of uncertainty coming from the overall scale. In this paper, we present a new method that addresses this task by employing two deep network stacks: one that makes a coarse global prediction based on the entire image, and another that refines this prediction locally. We also apply a scale-invariant error to help measure depth relations rather than scale. By leveraging the raw datasets as large sources of training data, our method achieves state-of-the-art results on both NYU Depth and KITTI, and matches detailed depth boundaries without the need for superpixelation.

## 1  Introduction

Estimating depth is an important component of understanding geometric relations within a scene. In turn, such relations help provide richer representations of objects and their environment, often leading to improvements in existing recognition tasks [18], as well as enabling many further applications such as 3D modeling [16, 6], physics and support models [18], robotics [4, 14], and potentially reasoning about occlusions.

While there is much prior work on estimating depth based on stereo images or motion [17], there has been relatively little on estimating depth from a *single* image. Yet the monocular case often arises in practice: Potential applications include better understandings of the many images distributed on the web and social media outlets, real estate listings, and shopping sites. These include many examples of both indoor and outdoor scenes.

There are likely several reasons why the monocular case has not yet been tackled to the same degree as the stereo one. Provided accurate image correspondences, depth can be recovered deterministically in the stereo case [5]. Thus, stereo depth estimation can be reduced to developing robust image point correspondences — which can often be found using local appearance features. By contrast, estimating depth from a single image requires the use of monocular depth cues such as line angles and perspective, object sizes, image position, and atmospheric effects. Furthermore, a global view of the scene may be needed to relate these effectively, whereas local disparity is sufficient for stereo.

Moreover, the task is inherently ambiguous, and a technically ill-posed problem: Given an image, an infinite number of possible world scenes may have produced it. Of course, most of these are physically implausible for real-world spaces, and thus the depth may still be predicted with considerable accuracy. At least one major ambiguity remains, though: the global scale. Although extreme cases (such as a normal room versus a dollhouse) do not exist in the data, moderate variations in room and furniture sizes are present. We address this using a *scale-invariant error* in addition to more

common scale-dependent errors. This focuses attention on the spatial relations within a scene rather than general scale, and is particularly apt for applications such as 3D modeling, where the model is often rescaled during postprocessing.

In this paper we present a new approach for estimating depth from a single image. We *directly regress on the depth* using a neural network with two components: one that first estimates the global structure of the scene, then a second that refines it using local information. The network is trained using a loss that explicitly accounts for depth relations between pixel locations, in addition to point-wise error. Our system achieves state-of-the art estimation rates on NYU Depth and KITTI, as well as improved qualitative outputs.

## 2   Related Work

Directly related to our work are several approaches that estimate depth from a single image. Saxena *et al.* [15] predict depth from a set of image features using linear regression and a MRF, and later extend their work into the Make3D [16] system for 3D model generation. However, the system relies on horizontal alignment of images, and suffers in less controlled settings. Hoiem *et al.* [6] do not predict depth explicitly, but instead categorize image regions into geometric structures (ground, sky, vertical), which they use to compose a simple 3D model of the scene.

More recently, Ladicky *et al.* [12] show how to integrate semantic object labels with monocular depth features to improve performance; however, they rely on handcrafted features and use super-pixels to segment the image. Karsch *et al.* [7] use a kNN transfer mechanism based on SIFT Flow [11] to estimate depths of static backgrounds from single images, which they augment with motion information to better estimate moving foreground subjects in videos. This can achieve better alignment, but requires the entire dataset to be available at runtime and performs expensive alignment procedures. By contrast, our method learns an easier-to-store set of network parameters, and can be applied to images in real-time.

More broadly, stereo depth estimation has been extensively investigated. Scharstein *et al.* [17] provide a survey and evaluation of many methods for 2-frame stereo correspondence, organized by matching, aggregation and optimization techniques. In a creative application of multiview stereo, Snavely *et al.* [20] match across views of many uncalibrated consumer photographs of the same scene to create accurate 3D reconstructions of common landmarks.

Machine learning techniques have also been applied in the stereo case, often obtaining better results while relaxing the need for careful camera alignment [8, 13, 21, 19]. Most relevant to this work is Konda *et al.* [8], who train a factored autoencoder on image patches to predict depth from stereo sequences; however, this relies on the local displacements provided by stereo.

There are also several hardware-based solutions for single-image depth estimation. Levin *et al.* [10] perform depth from defocus using a modified camera aperture, while the Kinect and Kinect v2 use active stereo and time-of-flight to capture depth. Our method makes indirect use of such sensors to provide ground truth depth targets during training; however, at test time our system is purely software-based, predicting depth from RGB images.

## 3   Approach

### 3.1   Model Architecture

Our network is made of two component stacks, shown in Fig. 1. A coarse-scale network first predicts the depth of the scene at a global level. This is then refined within local regions by a fine-scale network. Both stacks are applied to the original input, but in addition, the coarse network's output is passed to the fine network as additional first-layer image features. In this way, the local network can edit the global prediction to incorporate finer-scale details.

### 3.1.1   Global Coarse-Scale Network

The task of the coarse-scale network is to predict the overall depth map structure using a global view of the scene. The upper layers of this network are fully connected, and thus contain the entire image in their field of view. Similarly, the lower and middle layers are designed to combine information from different parts of the image through max-pooling operations to a small spatial dimension. In so doing, the network is able to integrate a global understanding of the full scene to predict the depth. Such an understanding is needed in the single-image case to make effective use of cues such

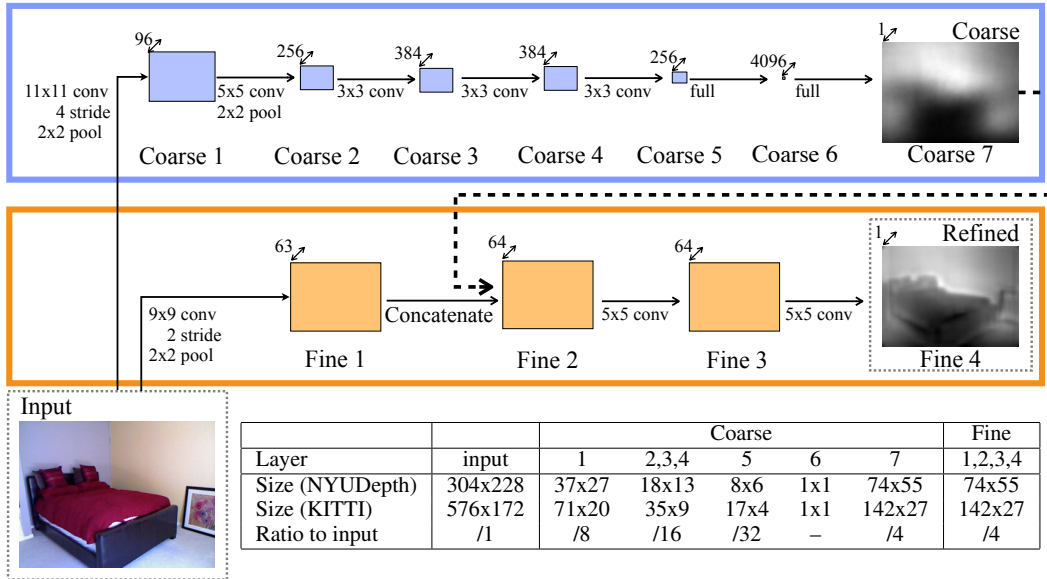

| | | | | Coarse | | | | Fine |
|---|---|---|---|---|---|---|---|---|
| Layer | input | 1 | 2,3,4 | 5 | 6 | 7 | | 1,2,3,4 |
| Size (NYUDepth) | 304x228 | 37x27 | 18x13 | 8x6 | 1x1 | 74x55 | | 74x55 |
| Size (KITTI) | 576x172 | 71x20 | 35x9 | 17x4 | 1x1 | 142x27 | | 142x27 |
| Ratio to input | /1 | /8 | /16 | /32 | – | /4 | | /4 |

Figure 1: Model architecture.

as vanishing points, object locations, and room alignment. A local view (as is commonly used for stereo matching) is insufficient to notice important features such as these.

As illustrated in Fig. 1, the global, coarse-scale network contains five feature extraction layers of convolution and max-pooling, followed by two fully connected layers. The input, feature map and output sizes are also given in Fig. 1. The final output is at $1/4$-resolution compared to the input (which is itself downsampled from the original dataset by a factor of 2), and corresponds to a center crop containing most of the input (as we describe later, we lose a small border area due to the first layer of the fine-scale network and image transformations).

Note that the spatial dimension of the output is larger than that of the topmost convolutional feature map. Rather than limiting the output to the feature map size and relying on hardcoded upsampling before passing the prediction to the fine network, we allow the top full layer to learn templates over the larger area (74x55 for NYU Depth). These are expected to be blurry, but will be better than the upsampled output of a 8x6 prediction (the top feature map size); essentially, we allow the network to learn its own upsampling based on the features. Sample output weights are shown in Fig. 2

All hidden layers use rectified linear units for activations, with the exception of the coarse output layer 7, which is linear. Dropout is applied to the fully-connected hidden layer 6. The convolutional layers (1-5) of the coarse-scale network are pretrained on the ImageNet classification task [1] — while developing the model, we found pretraining on ImageNet worked better than initializing randomly, although the difference was not very large[1].

### 3.1.2 Local Fine-Scale Network

After taking a global perspective to predict the coarse depth map, we make local refinements using a second, fine-scale network. The task of this component is to edit the coarse prediction it receives to align with local details such as object and wall edges. The fine-scale network stack consists of convolutional layers only, along with one pooling stage for the first layer edge features.

While the coarse network sees the entire scene, the field of view of an output unit in the fine network is 45x45 pixels of input. The convolutional layers are applied across feature maps at the target output size, allowing a relatively high-resolution output at $1/4$ the input scale.

More concretely, the coarse output is fed in as an additional low-level feature map. By design, the coarse prediction is the same spatial size as the output of the first fine-scale layer (after pooling),

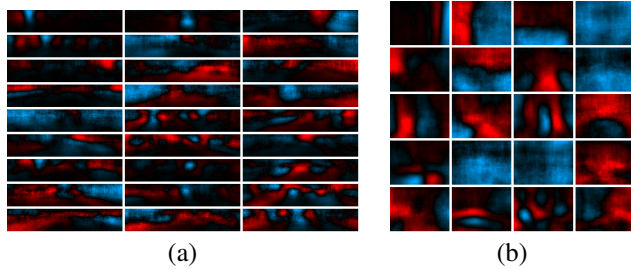

|       (a)       |       (b)       |

Figure 2: Weight vectors from layer Coarse 7 (coarse output), for (*a*) KITTI and (*b*) NYUDepth. Red is positive (farther) and blue is negative (closer); black is zero. Weights are selected uniformly and shown in descending order by $l_2$ norm. KITTI weights often show changes in depth on either side of the road. NYUDepth weights often show wall positions and doorways.

and we concatenate the two together (Fine 2 in Fig. 1). Subsequent layers maintain this size using zero-padded convolutions.

All hidden units use rectified linear activations. The last convolutional layer is linear, as it predicts the target depth. We train the coarse network first against the ground-truth targets, then train the fine-scale network keeping the coarse-scale output fixed (*i.e.* when training the fine network, we do not backpropagate through the coarse one).

### 3.2 Scale-Invariant Error

The global scale of a scene is a fundamental ambiguity in depth prediction. Indeed, much of the error accrued using current elementwise metrics may be explained simply by how well the mean depth is predicted. For example, Make3D trained on NYUDepth obtains 0.41 error using RMSE in log space (see Table 1). However, using an oracle to substitute the mean log depth of each prediction with the mean from the corresponding ground truth reduces the error to 0.33, a 20% relative improvement. Likewise, for our system, these error rates are 0.28 and 0.22, respectively. Thus, just finding the average scale of the scene accounts for a large fraction of the total error.

Motivated by this, we use a scale-invariant error to measure the relationships between points in the scene, irrespective of the absolute global scale. For a predicted depth map $y$ and ground truth $y^*$, each with $n$ pixels indexed by $i$, we define the *scale-invariant mean squared error* (in log space) as

$$D(y, y^*) \quad = \quad \frac{1}{2n} \sum_{i=1}^{n} (\log y_i - \log y_i^* + \alpha(y, y^*))^2, \tag{1}$$

where $\alpha(y, y^*) = \frac{1}{n} \sum_i (\log y_i^* - \log y_i)$ is the value of $\alpha$ that minimizes the error for a given $(y, y^*)$. For any prediction $y$, $e^\alpha$ is the scale that best aligns it to the ground truth. All scalar multiples of $y$ have the same error, hence the scale invariance.

Two additional ways to view this metric are provided by the following equivalent forms. Setting $d_i = \log y_i - \log y_i^*$ to be the difference between the prediction and ground truth at pixel $i$, we have

$$D(y, y^*) \quad = \quad \frac{1}{2n^2} \sum_{i,j} \left( (\log y_i - \log y_j) - (\log y_i^* - \log y_j^*) \right)^2 \tag{2}$$

$$= \quad \frac{1}{n} \sum_i d_i^2 - \frac{1}{n^2} \sum_{i,j} d_i d_j \quad = \quad \frac{1}{n} \sum_i d_i^2 - \frac{1}{n^2} \left( \sum_i d_i \right)^2 \tag{3}$$

Eqn. 2 expresses the error by comparing relationships *between pairs* of pixels $i, j$ in the output: to have low error, each pair of pixels in the prediction must differ in depth by an amount similar to that of the corresponding pair in the ground truth. Eqn. 3 relates the metric to the original $l_2$ error, but with an additional term, $-\frac{1}{n^2} \sum_{ij} d_i d_j$, that credits mistakes if they are in the same direction and penalizes them if they oppose. Thus, an imperfect prediction will have lower error when its mistakes are consistent with one another. The last part of Eqn. 3 rewrites this as a linear-time computation.

In addition to the scale-invariant error, we also measure the performance of our method according to several error metrics have been proposed in prior works, as described in Section 4.

### 3.3 Training Loss

In addition to performance evaluation, we also tried using the scale-invariant error as a training loss. Inspired by Eqn. 3, we set the per-sample training loss to

$$L(y, y^*) \;\;=\;\; \frac{1}{n}\sum_i d_i^2 - \frac{\lambda}{n^2}\left(\sum_i d_i\right)^2 \tag{4}$$

where $d_i = \log y_i - \log y_i^*$ and $\lambda \in [0,1]$. Note the output of the network is $\log y$; that is, the final linear layer predicts the log depth. Setting $\lambda = 0$ reduces to elementwise $l_2$, while $\lambda = 1$ is the scale-invariant error exactly. We use the average of these, *i.e.* $\lambda = 0.5$, finding that this produces good absolute-scale predictions while slightly improving qualitative output.

During training, most of the target depth maps will have some missing values, particularly near object boundaries, windows and specular surfaces. We deal with these simply by masking them out and evaluating the loss only on valid points, *i.e.* we replace $n$ in Eqn. 4 with the number of pixels that have a target depth, and perform the sums excluding pixels $i$ that have no depth value.

### 3.4 Data Augmentation

We augment the training data with random online transformations (values shown for NYUDepth) [2]:

- *Scale*: Input and target images are scaled by $s \in [1, 1.5]$, and the depths are divided by $s$.
- *Rotation*: Input and target are rotated by $r \in [-5, 5]$ degrees.
- *Translation*: Input and target are randomly cropped to the sizes indicated in Fig. 1.
- *Color*: Input values are multiplied globally by a random RGB value $c \in [0.8, 1.2]^3$.
- *Flips*: Input and target are horizontally flipped with 0.5 probability.

Note that image scaling and translation do not preserve the world-space geometry of the scene. This is easily corrected in the case of scaling by dividing the depth values by the scale $s$ (making the image $s$ times larger effectively moves the camera $s$ times closer). Although translations are not easily fixed (they effectively change the camera to be incompatible with the depth values), we found that the extra data they provided benefited the network even though the scenes they represent were slightly warped. The other transforms, flips and in-plane rotation, are geometry-preserving. At test time, we use a single center crop at scale 1.0 with no rotation or color transforms.

## 4 Experiments

We train our model on the raw versions both NYU Depth v2 [18] and KITTI [3]. The raw distributions contain many additional images collected from the same scenes as in the more commonly used small distributions, but with no preprocessing; in particular, points for which there is no depth value are left unfilled. However, our model's natural ability to handle such gaps as well as its demand for large training sets make these fitting sources of data.

### 4.1 NYU Depth

The NYU Depth dataset [18] is composed of 464 indoor scenes, taken as video sequences using a Microsoft Kinect camera. We use the official train/test split, using 249 scenes for training and 215 for testing, and construct our training set using the raw data for these scenes. RGB inputs are downsampled by half, from 640x480 to 320x240. Because the depth and RGB cameras operate at different variable frame rates, we associate each depth image with its closest RGB image in time, and throw away frames where one RGB image is associated with more than one depth (such a one-to-many mapping is not predictable). We use the camera projections provided with the dataset to align RGB and depth pairs; pixels with no depth value are left missing and are masked out. To remove many invalid regions caused by windows, open doorways and specular surfaces we also mask out depths equal to the minimum or maximum recorded for each image.

The training set has 120K unique images, which we shuffle into a list of 220K after evening the scene distribution (1200 per scene). We test on the 694-image NYU Depth v2 test set (with filled-in depth values). We train the coarse network for 2M samples using SGD with batches of size 32. We then hold it fixed and train the fine network for 1.5M samples (given outputs from the already-trained coarse one). Learning rates are: 0.001 for coarse convolutional layers 1-5, 0.1 for coarse full layers 6 and 7, 0.001 for fine layers 1 and 3, and 0.01 for fine layer 2. These ratios were found by trial-and-error on a validation set (folded back into the training set for our final evaluations), and the global scale of all the rates was tuned to a factor of 5. Momentum was 0.9. Training took 38h for the coarse network and 26h for fine, for a total of 2.6 days using a NVidia GTX Titan Black. Test prediction takes 0.33s per batch (0.01s/image).

## 4.2 KITTI

The KITTI dataset [3] is composed of several outdoor scenes captured while driving with car-mounted cameras and depth sensor. We use 56 scenes from the "city," "residential," and "road" categories of the raw data. These are split into 28 for training and 28 for testing. The RGB images are originally 1224x368, and downsampled by half to form the network inputs.

The depth for this dataset is sampled at irregularly spaced points, captured at different times using a rotating LIDAR scanner. When constructing the ground truth depths for training, there may be conflicting values; since the RGB cameras shoot when the scanner points forward, we resolve conflicts at each pixel by choosing the depth recorded closest to the RGB capture time. Depth is only provided within the bottom part of the RGB image, however we feed the entire image into our model to provide additional context to the global coarse-scale network (the fine network sees the bottom crop corresponding to the target area).

The training set has 800 images per scene. We exclude shots where the car is stationary (acceleration below a threshold) to avoid duplicates. Both left and right RGB cameras are used, but are treated as unassociated shots. The training set has 20K unique images, which we shuffle into a list of 40K (including duplicates) after evening the scene distribution. We train the coarse model first for 1.5M samples, then the fine model for 1M. Learning rates are the same as for NYU Depth. Training took took 30h for the coarse model and 14h for fine; test prediction takes 0.40s/batch (0.013s/image).

## 4.3 Baselines and Comparisons

We compare our method against Make3D trained on the same datasets, as well as the published results of other current methods [12, 7]. As an additional reference, we also compare to the mean depth image computed across the training set. We trained Make3D on KITTI using a subset of 700 images (25 per scene), as the system was unable to scale beyond this size. Depth targets were filled in using the colorization routine in the NYUDepth development kit. For NYUDepth, we used the common distribution training set of 795 images. We evaluate each method using several errors from prior works, as well as our scale-invariant metric:

| | |
|---|---|
| Threshold: % of $y_i$ s.t. $\max(\frac{y_i}{y_i^*}, \frac{y_i^*}{y_i}) = \delta < thr$ | RMSE (linear): $\sqrt{\frac{1}{|T|} \sum_{y \in T} \|y_i - y_i^*\|^2}$ |
| Abs Relative difference: $\frac{1}{|T|} \sum_{y \in T} |y - y^*|/y^*$ | RMSE (log): $\sqrt{\frac{1}{|T|} \sum_{y \in T} \|\log y_i - \log y_i^*\|^2}$ |
| Squared Relative difference: $\frac{1}{|T|} \sum_{y \in T} \|y - y^*\|^2/y^*$ | RMSE (log, scale-invariant): The error Eqn. 1 |

Note that the predictions from Make3D and our network correspond to slightly different center crops of the input. We compare them on the intersection of their regions, and upsample predictions to the full original input resolution using nearest-neighbor. Upsampling negligibly affects performance compared to downsampling the ground truth and evaluating at the output resolution. [3]

# 5 Results

## 5.1 NYU Depth

Results for NYU Depth dataset are provided in Table 1. As explained in Section 4.3, we compare against the data mean and Make3D as baselines, as well as Karsch *et al.* [7] and Ladicky *et al.* [12]. (Ladicky *et al.* uses a joint model which is trained using both depth and semantic labels). Our system achieves the best performance on all metrics, obtaining an average 35% relative gain compared to the runner-up. Note that our system is trained using the raw dataset, which contains many more example instances than the data used by other approaches, and is able to effectively leverage it to learn relevant features and their associations.

This dataset breaks many assumptions made by Make3D, particularly horizontal alignment of the ground plane; as a result, Make3D has relatively poor performance in this task. Importantly, our method improves over it on both scale-dependent and scale-invariant metrics, showing that our system is able to predict better relations as well as better means.

Qualitative results are shown on the left side of Fig. 4, sorted top-to-bottom by scale-invariant MSE. Although the fine-scale network does not improve in the error measurements, its effect is clearly visible in the depth maps — surface boundaries have sharper transitions, aligning to local details. However, some texture edges are sometimes also included. Fig. 3 compares Make3D as well as

| | Mean | Make3D | Ladicky&al | Karsch&al | Coarse | Coarse + Fine | |
|---|---|---|---|---|---|---|---|
| threshold $\delta < 1.25$ | 0.418 | 0.447 | 0.542 | – | **0.618** | 0.611 | higher |
| threshold $\delta < 1.25^2$ | 0.711 | 0.745 | 0.829 | – | **0.891** | 0.887 | is |
| threshold $\delta < 1.25^3$ | 0.874 | 0.897 | 0.940 | – | 0.969 | **0.971** | better |
| abs relative difference | 0.408 | 0.349 | – | 0.350 | 0.228 | **0.215** | |
| sqr relative difference | 0.581 | 0.492 | – | – | 0.223 | **0.212** | lower |
| RMSE (linear) | 1.244 | 1.214 | – | 1.2 | **0.871** | 0.907 | is |
| RMSE (log) | 0.430 | 0.409 | – | – | **0.283** | 0.285 | better |
| RMSE (log, scale inv.) | 0.304 | 0.325 | – | – | 0.221 | **0.219** | |

Table 1: Comparison on the NYUDepth dataset

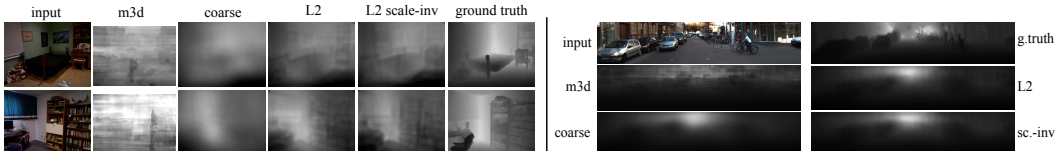

Figure 3: Qualitative comparison of Make3D, our method trained with $l_2$ loss ($\lambda = 0$), and our method trained with both $l_2$ and scale-invariant loss ($\lambda = 0.5$).

outputs from our network trained using losses with $\lambda = 0$ and $\lambda = 0.5$. While we did not observe numeric gains using $\lambda = 0.5$, it did produce slight qualitative improvements in more detailed areas.

## 5.2 KITTI

We next examine results on the KITTI driving dataset. Here, the Make3D baseline is well-suited to the dataset, being composed of horizontally aligned images, and achieves relatively good results. Still, our method improves over it on all metrics, by an average 31% relative gain. Just as importantly, there is a 25% gain in both the scale-dependent and scale-invariant RMSE errors, showing there is substantial improvement in the predicted structure. Again, the fine-scale network does not improve much over the coarse one in the error metrics, but differences between the two can be seen in the qualitative outputs.

The right side of Fig. 4 shows examples of predictions, again sorted by error. The fine-scale network produces sharper transitions here as well, particularly near the road edge. However, the changes are somewhat limited. This is likely caused by uncorrected alignment issues between the depth map and input in the training data, due to the rotating scanner setup. This dissociates edges from their true position, causing the network to average over their more random placements. Fig. 3 shows Make3D performing much better on this data, as expected, while using the scale-invariant error as a loss seems to have little effect in this case.

| | Mean | Make3D | Coarse | Coarse + Fine | |
|---|---|---|---|---|---|
| threshold $\delta < 1.25$ | 0.556 | 0.601 | 0.679 | **0.692** | higher |
| threshold $\delta < 1.25^2$ | 0.752 | 0.820 | 0.897 | **0.899** | is |
| threshold $\delta < 1.25^3$ | 0.870 | 0.926 | **0.967** | 0.967 | better |
| abs relative difference | 0.412 | 0.280 | 0.194 | **0.190** | |
| sqr relative difference | 5.712 | 3.012 | 1.531 | **1.515** | lower |
| RMSE (linear) | 9.635 | 8.734 | 7.216 | **7.156** | is |
| RMSE (log) | 0.444 | 0.361 | 0.273 | **0.270** | better |
| RMSE (log, scale inv.) | 0.359 | 0.327 | 0.248 | **0.246** | |

Table 2: Comparison on the KITTI dataset.

## 6 Discussion

Predicting depth estimates from a single image is a challenging task. Yet by combining information from both global and local views, it can be performed reasonably well. Our system accomplishes this through the use of two deep networks, one that estimates the global depth structure, and another that refines it locally at finer resolution. We achieve a new state-of-the-art on this task for NYU Depth and KITTI datasets, having effectively leveraged the full raw data distributions.

In future work, we plan to extend our method to incorporate further 3D geometry information, such as surface normals. Promising results in normal map prediction have been made by Fouhey *et al.* [2], and integrating them along with depth maps stands to improve overall performance [16]. We also hope to extend the depth maps to the full original input resolution by repeated application of successively finer-scaled local networks.

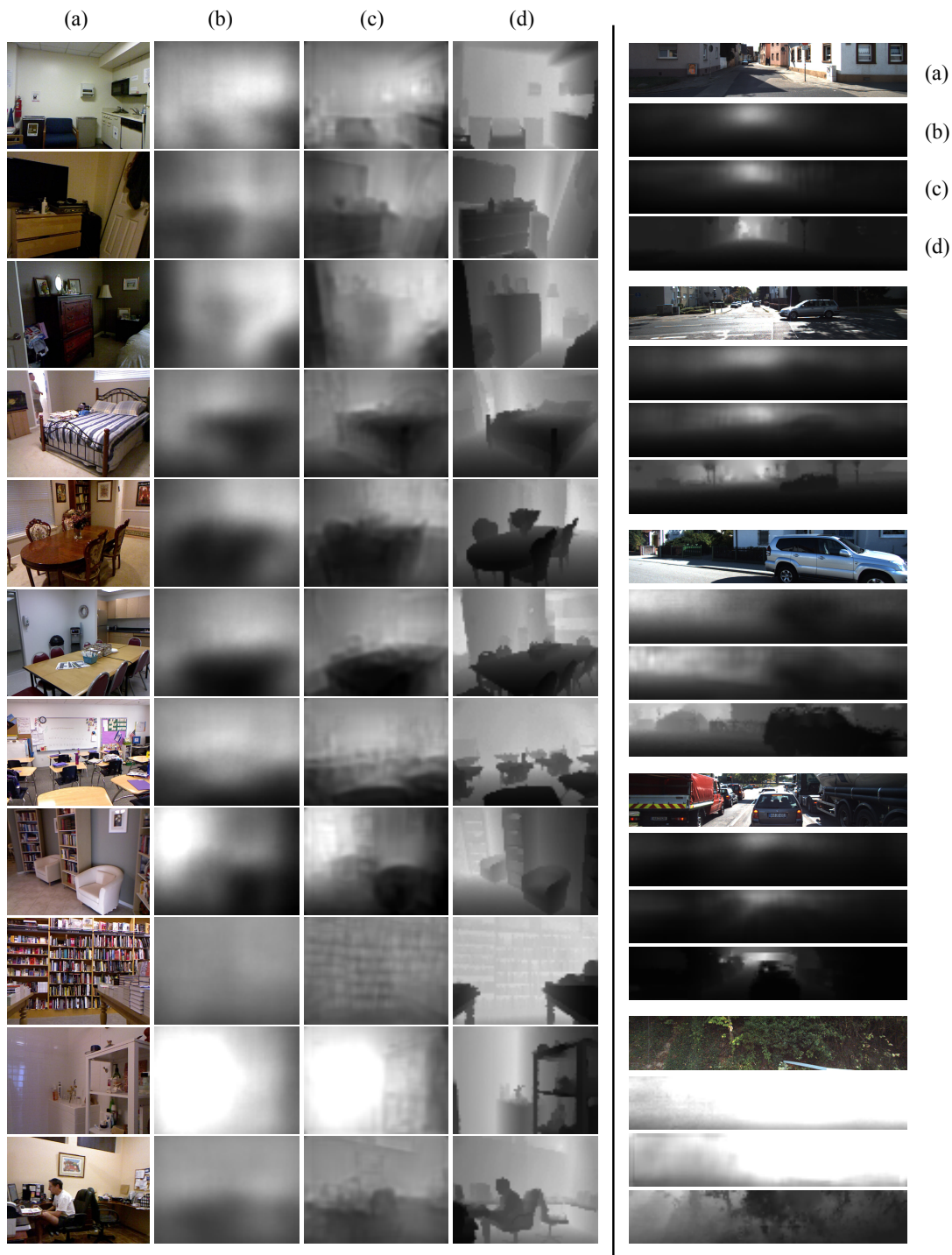

Figure 4: Example predictions from our algorithm. NYUDepth on left, KITTI on right. For each image, we show (a) input, (b) output of coarse network, (c) refined output of fine network, (d) ground truth. The fine scale network edits the coarse-scale input to better align with details such as object boundaries and wall edges. Examples are sorted from best (top) to worst (bottom).

## Acknowledgements

The authors are grateful for support from ONR #N00014-13-1-0646, NSF #1116923, #1149633 and Microsoft Research.

## Footnotes

[1]When pretraining, we stack two fully connected layers with 4096 - 4096 - 1000 output units each, with dropout applied to the two hidden layers, as in [9]. We train the network using random 224x224 crops from the center 256x256 region of each training image, rescaled so the shortest side has length 256. This model achieves a top-5 error rate of 18.1% on the ILSVRC2012 validation set, voting with 2 flips and 5 translations per image.

[2]For KITTI, $s \in [1, 1.2]$, and rotations are not performed (images are horizontal from the camera mount).

[3]On NYUDepth, log RMSE is 0.285 vs 0.286 for upsampling and downsampling, respectively, and scale-invariant RMSE is 0.219 vs 0.221. The intersection is 86% of the network region and 100% of Make3D for NYUDepth, and 100% of the network and 82% of Make3D for KITTI.

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
