[Supplementary Material]

# Depth Map Prediction from a Single Image using a Multi-Scale Deep Network: Supplementary Example Images

**David Eigen**
deigen@cs.nyu.edu

**Christian Puhrsch**
cpuhrsch@nyu.edu

**Rob Fergus**
fergus@cs.nyu.edu

The following pages contain additional comparison images.

Note this file uses substantial JPEG compression and there are noticeable image artifacts. Please see our project webpage for additional examples without compression:

http://www.cs.nyu.edu/~deigen/depth

# NYUDepth: Additional Examples

| input | m3d | coarse | fine (elem.) | fine | ground truth |
|---|---|---|---|---|---|

# KITTI: Additonal Examples

| input | m3d | coarse | fine (elem.) | fine | ground truth |
|-------|-----|--------|--------------|------|--------------|