[Reviews · NeurIPS 2014]

Submitted by Assigned_Reviewer_11

The authors present a new method for estimating the depth map of a scene using a single image. They use two CNNs: the first outputs a coarse depth prediction based on the entire image. This coarse depth prediction is used as the input, together with the original image, to the second CNN which refines the depth prediction locally. They also present and apply a scale-invariant error measure for the depth prediction. They demonstrate state of the art results on NYU Depth and KITTI datasets.

The paper is well written and clear, I believe researchers who would like to reproduce the results could do that using the details found in the paper. I also find that the problem they solve - estimating the depth of a scene using a single image, is relevant and interesting to the NIPS community.
However, much of the significance of the paper and the interest it raises is related to the two CNNs it uses - global and local. I would like to see better analysis of the importance of this two networks. It seems that quantitatively there isn't much of a difference to when using only the coarse CNN and both CNNs. How would the results look like if we would feed the second CNN with some other depth prediction which is not "coarse" (e.g. the output of Make3D), or if the first CNN will also be more local?

In the end of the first paragraph of 3.1.1 the authors explain the importance of global cues for depth estimation, such as vanishing points and room alignment. Is there a way to understand the patterns learned as such cues?

The fact that the choice of the lambda parameter of the loss doesn't give numeric gain but changes the quality of the prediction is a bit troubling (as the use of the second network) and suggests that the error measure used is not the most suited for this task.

I must say that when examining qualitatively the results presented both in the paper and in the supplementary material it seems that: (a) the depth estimate is still quite far from the ground truth. (b) on the KITTI dataset, it doesn't seem to me that the suggested architecture is better than Move3D.

I would like to see the running time and the computational resources needed for training and testing.

Minor comment:
Can it be that you forgot a factor 2 when moving from eq. 2 to eq. 3?
Summary: Well written paper about a subject with interest for the community. It lacks some better understanding of the results which makes it a borderline paper.

Submitted by Assigned_Reviewer_29

Summary:

This paper uses two deep network stacks to predict depth from a single image: the first network makes a coarse global prediction of depth, and the second network takes the input image and output of the first deep network to refine the prediction. The proposed algorithm achieves state-of-the-art accuracy on the datasets NYU Depth and KITTI.

Comments:

Existing approaches for depth map prediction from a single image have shortcomings. The Make3D system relies on several hand-engineered features and assumes horizontal alignment of images. The approach in [7] uses a kNN transfer mechanism and requires the entire dataset to be present at run-time. An important contribution of this paper is to propose the two deep network stacks: this is conceptually much simpler than the approaches above, is easier to implement, and makes it possible to apply the algorithm in real time once the two networks have been trained. Therefore, this work has the potential to have more impact than the existing techniques. The performance improvements are significant: relative gain over the runner up for NYU Depth dataset is 34%.

Questions about the model architecture and training:
- How does fixing the weights of the first 5 layers of the coarse network to the parameters trained using imageNet affect performance (i.e. only the last two fully connected layers are trained)?
What do the filters trained in the first layer of the second deep network look like? Are they Gabor, or different?
- What is the impact of having only 1 layer after the Coarse 7 layer is concatenated with the Fine 1 layer (i.e. no Fine 3 layer)?
- What are the computational resources that were used to train the network? What is the distribution of training time for the two networks?

Training the two networks one after the next feels unsatisfactory. Is it possible to derive an architecture where a single network can be trained in 1 pass, and achieve comparable results?
Summary: This paper proposes two deep network stacks for the problem of estimating depth from a single image. The proposed algorithm significantly outperforms state-of-the-art on two common benchmarks.

Submitted by Assigned_Reviewer_36

This paper presents a deep-network approach to the problem of depth estimation from a single image. Rather than applying a deep network to the problem in a trial-and-error manner, a multi-scale network architecture is proposed. The authors integrate both local information and global information by employing one network stack for coarse estimation, and the other network stack to refine the result for detail estimation. The paper is overall well-written, and the experiments based on popular datasets show promising results. The supplementary file provides additional experimental results.
Summary: This paper applies deep networks to an interesting problem: depth estimation from a single image. The proposed architecture is innovative, as it integrates global analysis and local analysis, and performs well with popular datasets.
Author Feedback
Author rebuttal: We thank all the reviewers for their comments and suggestions. We note all reviewers agree that the paper addresses a problem of interest, proposes an innovative method, achieves good results, and is well-written. Responses to individual comments are below:

R11
=========

"It seems that quantitatively there isn't much of a difference to when using only the coarse CNN and both CNNs."

While the quantitative measurements do not show much difference between the two networks' predictions, the qualitative difference is significant, with the combined network clearly improving over just the coarse one. We are continuing to investigate more improved error metrics, which may help address this in the future as well.

---

"analysis of the importance of this two networks ... How would the results look like if we would feed the second CNN with some other depth prediction ..., or if the first CNN will also be more local?"

Running the fine-scale network alone (without a coarse input) is unable to capture the depths provided by the coarse-scale prediction, and produces results with fine-scale variations, but little global structure. Both networks are necessary to obtain good results.

Using a fine-scale convnet to refine the output from another algorithm such as Make3D may indeed yield some improvement over the original predictions. However, it's highly doubtful this would beat our overall method, since the coarse network already outperforms Make3D on its own.

---

"Is there a way to understand the patterns learned"

We experimented a little with back-projection visualization methods, but did not find any silver bullets with this yet. We agree this is interesting and are continuing to investigate it, but also feel the paper offers a new perspective on an interesting problem.

---

"suggests that the error measure used is not the most suited for this task"

The error measure aligns to both an intuitive understanding of the task (predicting depth at each pixel and depth relations) and builds on prior works. We tried many error measures during development, and feel we made some incremental progress in this area; the combination of coarse and fine scales is our larger contribution. We do think further improved metrics are a good avenue for future work, however.

---

"the depth estimate is still quite far from the ground truth"

Although there is still room for improvement, the results are much better than previous approaches, particularly on NYUDepth. We feel this is both a significant advance and promising for future research.

---

"qualitatively ... on the KITTI dataset, it doesn't seem ... the suggested architecture is better than [Make3D]"

There are numerous examples in the supplement where our method is clearly better, especially on the last page (second page of KITTI examples); in particular rows 1, 2, 7-9 (l.61-64), 14 (l.69), 16 (l.72), 24 (l.81), 33 (l.90) of this page.

Make3D performs reasonably well for KITTI, since it is composed of horizontally-aligned street scenes. Nevertheless, Table 2 shows our approach clearly outperforming Make3D on KITTI.

---

"running time and the computational resources needed for training and testing"

For NYUDepth, training took 38h for the coarse network plus 26h for the fine network, for a total of 2.6 days; KITTI took 30h for coarse and 14h for fine. Testing takes 0.01s/image (0.33s/batch of 32 for NYUD; 0.40s/batch for KITTI). We used a GTX TITAN Black. We will include this in the final version.

---

"factor 2 when moving from eq. 2 to eq. 3"

Yes, thank you for the correction. (Eqn 2) = 2*(Eqn 3).

---

R29
=========

"How does fixing the weights of the first 5 layers of the coarse network to ... imageNet affect performance"

Fixing these weights degrades performance 7% relative for NYUD and 6% for KITTI, compared to backpropagating.

---

"What do the filters trained in the first layer of the second deep network look like?"

They are gabors.

---

"impact of having only 1 layer after the Coarse 7 layer is concatenated"

We were able to get a small gain with one additional layer here in subsequent experiments, although it is not large, about 1% relative.

---

"computational resources that were used to train the network?"

Please see response to same question by R11 above.

---

"derive an architecture where a single network can be trained in 1 pass, and achieve comparable results?"

This is an interesting question we have not yet examined in depth, but will try to explore more. Other architectures that we tried (some of which could be trained in a single pass) did not obtain as good results, both quantitatively and qualitatively.

R36
=========

We did not see any specific questions from R36, but appreciate and thank them for his/her review.